# Rethinking RoI Strategy in Interactive 3D Segmentation for Medical Images

Ziyu Zhang[1*][0000−0003−1423−7036], Yi Yu[2*][0000−0002−9841−4687], and Yuan Xue[2][0000−0002−5390−9037]

[1] Medical School, Nanjing University, Nanjing, 210096 China
[2] College of Medicine, The Ohio State University, Columbus, OH 43210 USA
Yuan.Xue@osumc.edu

**Abstract.** 3D biomedical image segmentation is a critical technology for clinical diagnostics, surgical planning, and disease analysis. While foundation models such as SAM and its medical derivatives have achieved remarkable progress, their adaptation to volumetric images remains limited, particularly in terms of diverse imaging modalities and the need for efficient user interaction. To advance research in this field, *CVPR 2025 Interactive 3D Biomedical Image Segmentation Challenge* was established. We propose DCM (DualClickMed) as a solution to this challenge, with a dual-expert architecture featuring both global and local Region-of-Interest (RoI) strategies. The global-RoI expert provides comprehensive anatomical context by processing the entire organ based on user prompts, while the local-RoI expert focuses on high-resolution patches centered on specific user clicks, enabling precise segmentation of fine structures. We further introduce tailored prompt simulation strategies for each expert, closely mimicking real-world interactive behaviors during training. Extensive experiments on challenge dataset covering five modalities demonstrate that our approach outperforms baselines, with final DSC scores of 0.8533 (CT), 0.6880 (MRI), 0.6003 (Microscopy), 0.7864 (PET), and 0.9385 (Ultrasound), achieving significant improvements in both region overlap and boundary accuracy metrics.

**Keywords:** Interactive 3D segmentation · Region-of-Interest strategy · Medical image analysis.

## 1  Introduction

3D biomedical image segmentation has become a critical technology for clinical diagnostics, surgical planning, and quantitative disease analysis. While foundation models like SAM (Segment Anything Model) have revolutionized 2D natural image segmentation through promptable architectures trained on billion-scale datasets [7], their application to volumetric medical imaging faces three key challenges: **1)** Medical images capture intricate structures across multiple scales, from tiny blood vessels to entire organs, requiring sophisticated analysis

---

⋆ Contributed equally.

[5]. **2)** Diverse modalities in medical imaging (CT, MRI, PET, Ultrasound, and Microscopy) and limited labeled data make learning effective representations particularly difficult [9,4]. **3)** Clinical workflows require interactive refinement capabilities where human expertise can efficiently correct segmentation errors, which is underdeveloped in current foundation models [14].

Recent research has attempted to bridge this gap. For instance, MedSAM [9] and MedSAM2 [11] perform domain-specific fine-tuning and sequence modeling, whereas SegVol [3] and VISTA3D [5] introduce multi-scale and unified 3D segmentation strategies. Some efforts such as One-Prompt [16] seek to design universal interactive prompts for diverse medical images.

Despite their promise, existing interactive segmentation methods face two critical shortcomings: **1)** While foundation models have shown success in medical segmentation, their interactive adaptation suffers from suboptimal RoI handling— where standard cropping/resizing discards crucial spatial context. **2)** Their prompt simulation strategies do not faithfully replicate real-world user interactions or effectively utilize anatomical context, resulting in poor performance for ambiguous boundaries and complex structures [15].

Motivated by these limitations, we propose DCM (DualClickMed), an interactive segmentation framework incorporating two RoI strategies, with a global-RoI expert for holistic anatomical understanding and a local-RoI expert for fine-grained refinement. Complemented by realistic prompt simulation strategies, we achieve better training-application alignment and superior accuracy.

**Contributions. 1)** We propose a dual-expert medical segmentation framework featuring two complementary RoI modes: global and local, enabling both holistic organ context modeling and fine-grained local refinement in interactive 3D segmentation. **2)** We introduce two tailored interaction simulation strategies for training the global-RoI and local-RoI experts, making the training process more aligned with real-world user behaviors and improving model robustness to various prompt types. **3)** Extensive experiments on multiple challenging modalities demonstrate that our method significantly advances the state-of-the-art, consistently outperforming strong baselines in both region overlap and boundary accuracy metrics. The source code will be made publicly available.

## 2    Method

This section presents our methodological contributions in three parts: First, we introduce the architecture of our dual-expert segmentation framework (Sec. 2.1). Next, we detail our RoI extraction pipelines that serve our global and local experts (Sec. 2.2). Finally, we present the distinct interaction simulation strategies developed for training each expert model (Sec. 2.3).

### 2.1    Network Architecture

Our proposed framework introduces a dual-expert architecture for medical image segmentation that combines global and local processing strategies. As illustrated

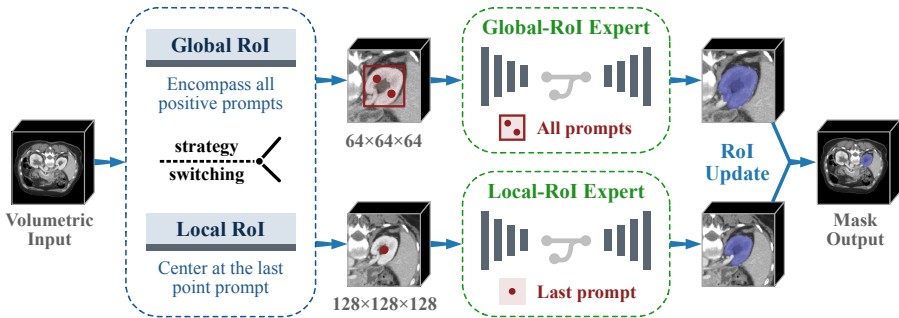

**Fig. 1.** Network architecture. Our method contains two expert models, one for global RoI and the other for local RoI: **1)** Global RoI: The input image is cropped and resampled based on the enclosing box of all prompts. **2)** Local RoI: The input image is cropped with its center at the last point prompt.

in Figure 1, the system employs two expert models working in tandem: one handling global RoI for complete anatomical structures, and another focusing on local RoI patches. This design addresses the varying requirements of different clinical segmentation scenarios through complementary approaches.

**Global RoI.** The global-RoI component processes entire organ structures (e.g. kidneys) based on interactive user prompts. Built upon a modified SAM-Med3D [15] architecture with Vision Transformer [2] backbone, we reduce the patch embedding size from $16 \times 16 \times 16$ to $8 \times 8 \times 8$ pixels. This modification also lowers the input resolution from $128 \times 128 \times 128$ to $64 \times 64 \times 64$, significantly improving computational efficiency for whole-organ analysis. Meanwhile, as all interactive points are generally within the global RoI, we input all the points into the prompt encoder, which is proven better in our ablation study.

The global approach provides comprehensive anatomical context but faces inherent limitations in certain clinical scenarios: **1)** When the input prompts only contain one point, it is not sufficient to extract global RoI. **2)** In some cases with thin structure and large range (e.g. vessels), the resolution of global RoI is not enough to do the segmentation.

**Local RoI.** To address the constraints of global processing, we incorporate a local-RoI expert based on a streamlined SegResNet [12]. This component focuses on high-resolution patch centered around user click points. It is adapted from VISTA3D [5], omitting the auto head to better align with interactive segmentation workflows. The local expert excels in scenarios where global processing proves inadequate, particularly when handling single-point prompts or segmenting fine anatomical details (e.g. vessels) that demand higher resolution. We use a simple rule to switch strategies. When only point prompts are provided, we pass the input image to local RoI, otherwise the global one is used.

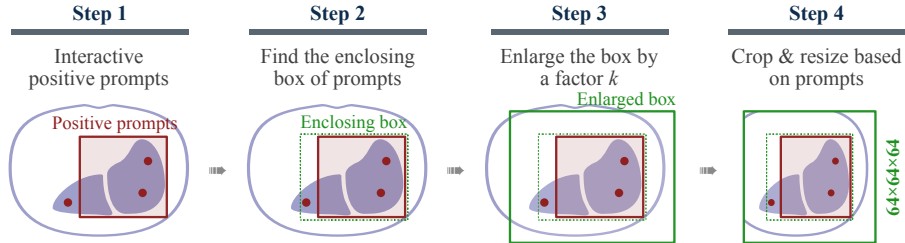

**Fig. 2.** Global RoI extractor. We propose to crop and resize the 3D medical image by finding the minimal enclosing box of all the interactive positive prompts.

It can be seen that this paper does involve a lot of modifications in terms of the network details, with both global and local experts based on existing models. Therefore, SAM-Med3D and VISTA3D are employed as the major counterparts. By validating against these two foundational models, we demonstrate how our devised mechanism improve the accuracy.

### 2.2   Global and Local RoI Extractors

The zoom strategy is a long-standing problem due to the high resolution of 3D medical image. When it comes to interactive segmentation, existing methods generally follow two strategies: **1)** Updating with siding windows (e.g. nnInteractive [4]). **2)** Updating around the prompt (e.g. VISTA3D [5]).

In this paper, we propose a novel strategy, updating RoI based on the minimal enclosing box of all the interactive positive prompts, as shown in Figure 2. To extract the global RoI, we first find the enclosing box of all positive prompts. Next, we enlarge the enclosing box, with its width and height multiplied by a factor $k$. The factor is set to 1.8 based on the observation that it is generally sufficient to include the whole organ. Finally, this enlarged box is cropped from original image and resized to $64 \times 64 \times 64$. The strategy has advantages in two folds: **1)** The global RoI aligns with the spatial extension of the target organ, making it less susceptible to resolution variation. **2)** No matter for small or large organs, the segmentation can be done in one inference, which is efficient.

On the other hand, the local RoI extractor we use is much simpler. It crops the $128 \times 128 \times 128$ patch centered around the last user click points, which is the same as SAM-Med3D [15] and VISTA3D [5].

### 2.3   Interaction Simulation in Training

The interaction simulation is also a crucial but underexplored part in interactive segmentation. Methods like SAM-Med3D and VISTA3D generate point prompts by randomly choose from the false prediction region. Such strategy is intuitive, but quite different from human behavior, making the training less efficient.

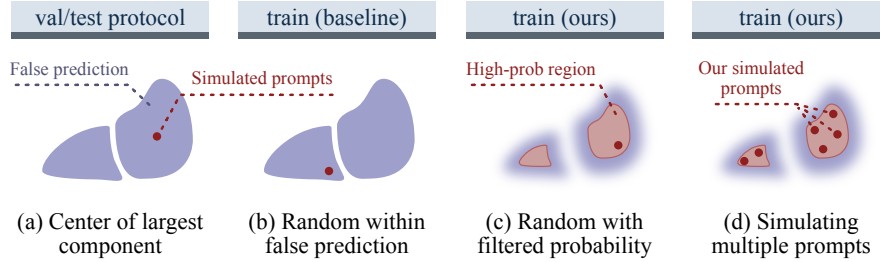

**Fig. 3.** Point prompt simulation for training global RoI expert. We propose two novel designs: **1)** Probability-based prompt simulation similar to human behavior. **2)** Simulating multiple prompts in each interaction for efficient learning.

We seek to interaction simulation strategies that are: **1)** Similar to human behavior. **2)** Fast enough for training. Two strategies are thus proposed, one for training the global-RoI expert, the other for the local one.

**Boundary-attenuated simulation for global RoI.** Let $M \in \{0,1\}^{D \times H \times W}$ denote the binary false prediction map from the previous segmentation output. A naive random sampling of interaction points from $M$ would disproportionately select surface voxels due to their higher spatial frequency. To better emulate human annotation behavior, we develop a sampling strategy as illustrated in Figure 3(c). First, we apply 3D Gaussian filter with $\sigma = 1$ to obtain a weighted map $G = \mathcal{N} * M$, where $\mathcal{N}$ represents the Gaussian kernel. This operation attenuates values at boundary regions while preserving interior voxel intensities. Next, we identify high-probability sampling regions by computing $H = \text{top}_k(G)$ where $k = 64$, selecting voxels with the highest activation values. Finally, simulated interaction points are uniformly sampled from this refined region $H$, effectively shifting the sampling distribution away from superficial areas toward more anatomically meaningful interior locations. This strategy improves the score by 1.79% according to our ablation study.

**Training efficiency optimization for global RoI.** While local RoIs require individual cropping per interaction point, the global RoI's whole-organ coverage enables simultaneous processing of multiple prompts within a single crop. As illustrated in Figure 3(d), we employ a batch processing approach where each image encoder forward pass is coupled with 30 prompt iterations (6 points $\times$ 5 interactions). This design is based on the observation that the prompt encoder and mask decoder operate significantly faster than the image encoder, making the additional prompt iterations minimally impact the overall training time. All 30 predicted masks are evaluated against ground truth annotations, with the average loss computed across all outputs driving the gradient update. Our ablation study shows this strategy improve the score by 0.38%.

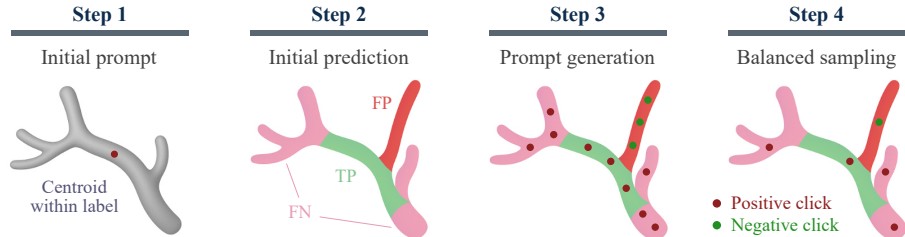

**Fig. 4.** Point prompt simulation for training local RoI expert. We propose to generate prompts for false positive, false negative, and confused region separately, and balance the proportion among them.

**Balanced prompt sampling for local RoI.** The local-RoI expert processes fixed-size $128 \times 128 \times 128$ patches, where foreground-background imbalance typically skews prompt sampling toward false positives over false negatives. To address this bias, we introduce a stratified prompt simulation strategy (see Figure 4) that explicitly balances prompt types during training. After generating an initial prediction, we categorize voxels into three error regions: **1)** FP (over-segmentation), **2)** FN (under-segmentation), and **3)** CR (confused boundary region where prediction probabilities fall near the 0.5 decision threshold). We then sample corrective prompts according to a probability distribution $(P_{FP} = 0.1, P_{FN} = 0.5, P_{CR} = 0.4)$, up-weighting FN and boundary regions to counteract their natural under-sampling. Ablation studies demonstrate that this strategy yields 1.50% improvement in segmentation accuracy by ensuring more representative prompt simulation during training.

### 2.4   Loss Functions

We use the summation between Dice loss and focal loss because compound loss functions have been proven to be robust in various medical image segmentation tasks [8]. In this work, the loss function design is inherited from existing work SAM-Med3D [15] and VISTA3D [5] without much modification.

## 3   Experiments

### 3.1   Dataset and Evaluation Metrics

The development set is an extension of the CVPR 2024 MedSAM on Laptop Challenge [10], including more 3D cases from public datasets[3] and covering commonly used 3D modalities, such as Computed Tomography (CT), Magnetic Resonance Imaging (MRI), Positron Emission Tomography (PET), Ultrasound, and Microscopy images. The hidden testing set is created by a community effort

---

[3] A complete list is available at https://medsam-datasetlist.github.io/

**Table 1.** Development and experimental environments.

| System | Red Hat Enterprise Linux 9 |
|---|---|
| CPU | AMD EPYC 7H12 CPU |
| RAM | $8 \times 60$GB |
| GPU | $8 \times$ NVIDIA A100 40GB |
| CUDA version | 12.6 |
| Programming language | Python 3.10 |
| Deep learning framework | PyTorch 2.6.0, Torchvision 0.21.0 |

where all the cases are unpublished. The annotations are either provided by the data contributors or annotated by the challenge organizer with 3D Slicer [6] and MedSAM2 [11]. In addition to using all training cases, the challenge contains a coreset track, where participants can select 10% of the total training cases for model development.

For each iterative segmentation, the evaluation metrics include Dice Similarity Coefficient (DSC) and Normalized Surface Distance (NSD) to evaluate the segmentation region overlap and boundary distance, respectively. The final metrics used for the ranking are:

- DSC_AUC and NSD_AUC Scores: AUC (Area Under the Curve) for DSC and NSD is used to measure cumulative improvement with interactions. The AUC quantifies the cumulative performance improvement over the five click predictions, providing a holistic view of the segmentation refinement process. It is computed only over the click predictions without considering the initial bounding box prediction as it is optional.
- Final DSC and NSD Scores after all refinements, indicating the model's final segmentation performance.

In addition, the algorithm runtime will be limited to 90 seconds per class. Exceeding this limit will lead to all DSC and NSD metrics being set to 0 for that test case.

### 3.2 Implementation Details

**Preprocessing.** Following the practice in MedSAM [9], all images were processed to npz format with an intensity range of $[0, 255]$. Specifically, for CT images, we initially normalized the Hounsfield units using typical window width and level values: soft tissues (W:400, L:40), lung (W:1500, L:-160), brain (W:80, L:40), and bone (W:1800, L:400). Subsequently, the intensity values were rescaled to the range of $[0, 255]$. For other images, we clipped the intensity values to the range between the 0.5th and 99.5th percentiles before rescaling them to the range of $[0, 255]$. If the original intensity range is already in $[0, 255]$, no preprocessing was applied.

**Table 2.** Training protocols for Global ROI and Local ROI.

|  | **Global ROI** | **Local ROI** |
|---|---|---|
| Pre-trained Model | SAM-Med3D | VISTA3D |
| Batch size | 4 | 1 |
| Patch size | 64×64×64 | 128×128×128 |
| Total epochs | 144 | 20 |
| Optimizer | AdamW | AdamW |
| Initial learning rate (lr) | $8 \times 10^{-4}$ | $2 \times 10^{-6}$ |
| Lr decay schedule | MultiStepLR | WarmupCosine |
| Training time | 20 hours | 30 hours |
| Number of model parameters | 100.51 M | 217 M |
| Number of flops | 150 G | 2026.66 G |

**Environment settings.** The environments used for development and experiments are summarized in Table 1. While our experiments were conducted using a distributed training setup with 8 GPUs (each with 18 CPU cores and 60GB memory), we emphasize that the proposed method maintains full functionality and can be effectively trained on a single GPU configuration.

**Training protocols.** The detail of training protocals for Global and Local ROI are summarized in Table 2. To improve the generalization ability of our model, we adopted comprehensive data augmentation strategies for both global and local RoI branches. For global-RoI, we employed the RandomAffine transformation from the TorchIO library [13], with scale factors of [0.9, 1.1] along the z-axis and [0.5, 1.5] along the x and y axes, as well as translation factor of 4 for all spatial axes. For local-RoI, the augmentation strategy is based on the MONAI framework [1], including random cropping guided by label classes, random intensity scaling and shifting, random additive Gaussian noise, random flipping along all spatial axes, and random 90-degree rotations. These augmentations are designed to simulate a wide range of imaging conditions and anatomical variations, effectively reducing overfitting and enhancing the robustness of our model.

For model selection, we adopt a straightforward strategy by using the final checkpoint obtained at the end of training. This "last checkpoint" policy allows us to fully utilize all training iterations and simplifies the model selection process without relying on additional validation-based early stopping.

## 4    Results and Discussion

### 4.1    Quantitative Results on Validation Set

**Coreset track.** Table 3 summarizes the quantitative performance of our method and three SOTA baselines (SAM-Med3D [15], VISTA3D [5], and SegVol [3]) on five imaging modalities: CT, MRI, Microscopy, PET, and Ultrasound. Our approach consistently outperforms all baselines in both DSC and NSD metrics. The

**Table 3.** Quantitative evaluation results of the validation set on the **coreset track**.

| Modality | Methods | DSC AUC | NSD AUC | DSC Final | NSD Final |
|---|---|---|---|---|---|
| CT | SAM-Med3D | 2.2408 | 2.2213 | 0.5590 | 0.5558 |
| | VISTA3D | 2.7975 | 2.8155 | 0.7147 | 0.7243 |
| | SegVol | 2.8987 | 3.0373 | 0.7247 | 0.7593 |
| | DCM (ours) | 3.3826 | 3.4891 | 0.8533 | 0.8818 |
| MRI | SAM-Med3D | 1.5191 | 1.5195 | 0.3895 | 0.3956 |
| | VISTA3D | 2.2901 | 2.5783 | 0.5777 | 0.6479 |
| | SegVol | 1.1131 | 1.3137 | 0.2783 | 0.3284 |
| | DCM (ours) | 2.7423 | 3.0951 | 0.6880 | 0.7738 |
| Microscopy | SAM-Med3D | 0.3042 | 0.0169 | 0.0768 | 0.0042 |
| | VISTA3D | 1.7183 | 2.7084 | 0.4455 | 0.6931 |
| | SegVol | 2.0355 | 3.4730 | 0.5089 | 0.8682 |
| | DCM (ours) | 2.3437 | 3.0661 | 0.6003 | 0.7750 |
| PET | SAM-Med3D | 2.1304 | 1.8150 | 0.5344 | 0.4560 |
| | VISTA3D | 2.3878 | 2.0984 | 0.6123 | 0.5430 |
| | SegVol | 2.9683 | 2.8563 | 0.7421 | 0.7141 |
| | DCM (ours) | 3.0990 | 2.9493 | 0.7864 | 0.7539 |
| Ultrasound | SAM-Med3D | 1.4347 | 1.7956 | 0.3841 | 0.5090 |
| | VISTA3D | 2.5803 | 2.5886 | 0.7074 | 0.7174 |
| | SegVol | 1.2325 | 1.7881 | 0.3081 | 0.4470 |
| | DCM (ours) | 3.7269 | 3.7637 | 0.9385 | 0.9519 |

improvements are particularly significant in the CT and Ultrasound modalities. Our method achieves a DSC of 0.8533 for CT and 0.9385 for Ultrasound, outperforming VISTA3D (0.7147 for CT, 0.7074 for Ultrasound) and SegVol (0.7247 for CT, 0.3081 for Ultrasound). For Ultrasound, the gain over the best baseline (VISTA3D) exceeds 23% in DSC, highlighting our model's superiority in handling low-quality images. On MRI and Microscopy, which are challenging due to noise and complex textures, our method also achieves clear improvements. For example, in Microscopy, our model attains a DSC of 0.6003, outperforming the best baseline (SegVol, 0.5089) by approximately 9%. Across all modalities, our consistent gains in both DSC and NSD metrics demonstrate the robustness and generalizability of our approach, as well as its effectiveness in accurately segmenting both high-contrast and low-quality medical images.

**All-data track.** Table 4 summarizes the quantitative evaluation results on the all-data track across five imaging modalities. Consistent with the results observed in the coreset track, our method achieves competitive or superior performance compared with the baselines (VISTA3D [15], SegVol [3], and nnInteractive [4]) in most modalities. Notably, in the Ultrasound modality, our approach attains the highest NSD (0.9440) and DSC (0.9299), outperforming all baselines by a clear margin (the best baseline, nnInteractive, achieves NSD of 0.8494 and DSC of 0.8547). For CT, our model also delivers strong results, with a NSD of 0.8797 and DSC of 0.8462, which is competitive with nnInteractive (NSD 0.9165,

**Table 4.** Quantitative evaluation results of the validation set on the **all-data track**.

| Modality | Methods | DSC AUC | NSD AUC | DSC Final | NSD Final |
|---|---|---|---|---|---|
| CT | SAM-Med3D | 2.2615 | 2.1533 | 0.5676 | 0.5421 |
| | VISTA3D | 3.1689 | 3.2652 | 0.8041 | 0.8344 |
| | SegVol | 2.9860 | 3.1191 | 0.7465 | 0.7798 |
| | nnInteractive | 3.4337 | 3.5743 | 0.8764 | 0.9165 |
| | DCM (ours) | 3.3461 | 3.4719 | 0.8462 | 0.8797 |
| MRI | SAM-Med3D | 1.6351 | 1.6106 | 0.4208 | 0.4193 |
| | VISTA3D | 2.5895 | 2.9683 | 0.6545 | 0.7493 |
| | SegVol | 1.2720 | 1.4629 | 0.3180 | 0.3657 |
| | nnInteractive | 2.6975 | 3.0292 | 0.7302 | 0.8227 |
| | DCM (ours) | 2.7133 | 3.0852 | 0.6809 | 0.7714 |
| Microscopy | SAM-Med3D | 0.3041 | 0.0168 | 0.0768 | 0.0042 |
| | VISTA3D | 2.0229 | 3.0150 | 0.5286 | 0.7701 |
| | SegVol | 2.2851 | 3.5661 | 0.5713 | 0.8915 |
| | nnInteractive | 3.0801 | 3.9027 | 0.7836 | 0.9813 |
| | DCM (ours) | 2.2917 | 3.0618 | 0.5871 | 0.7743 |
| PET | SAM-Med3D | 1.2879 | 0.7779 | 0.3219 | 0.1945 |
| | VISTA3D | 2.6398 | 2.3998 | 0.6779 | 0.6227 |
| | SegVol | 3.0225 | 2.9132 | 0.7556 | 0.7283 |
| | nnInteractive | 3.1877 | 3.0722 | 0.8156 | 0.7915 |
| | DCM (ours) | 3.0188 | 2.8778 | 0.7691 | 0.7440 |
| Ultrasound | SAM-Med3D | 1.7246 | 2.1188 | 0.4613 | 0.5597 |
| | VISTA3D | 2.8655 | 2.8441 | 0.8105 | 0.8079 |
| | SegVol | 3.4116 | 3.4167 | 0.8529 | 0.8542 |
| | nnInteractive | 3.3481 | 3.3236 | 0.8547 | 0.8494 |
| | DCM (ours) | 3.6741 | 3.7096 | 0.9299 | 0.9440 |

DSC 0.8764) and clearly surpasses VISTA3D and SegVol. For MRI and PET, our method achieves results comparable to the strongest baselines, demonstrating robust and generalizable performance across different imaging scenarios. In Microscopy, while nnInteractive achieves the best DSC (0.7836), our approach yields a DSC of 0.5871, outperforming VISTA3D and SAM-Med3D, and remaining competitive with SegVol. Overall, these results—together with the consistent gains observed in the coreset track—highlight the effectiveness and versatility of our dual-expert framework for large-scale 3D medical image segmentation.

### 4.2   Qualitative Results on Validation Set

Figure 5 illustrates representative successful segmentation cases across the five imaging modalities. For each modality, we present the input image, ground truth annotation, segmentation results from VISTA3D, and results from our method. Visually, our approach yields segmentations that are much closer to the ground truth compared to the baseline, particularly in accurately capturing fine anatomical structures and clear object boundaries. The improvements are especially noticeable in challenging modalities such as Ultrasound and Microscopy, where

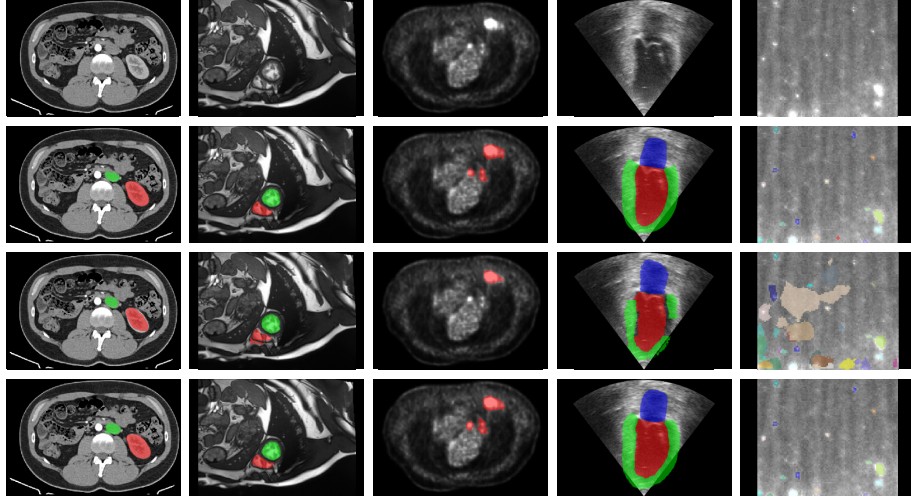

**Fig. 5.** Successful cases on five modalities. Top-to-bottom: Input images, Ground truths, Results of VISTA3D, and Results of ours. Left-to-right: CT, MRI, PET, Ultrasound, and Microscopy.

boundary definition is often difficult due to noise or low contrast. These qualitative results demonstrate the robustness and effectiveness of our method across diverse imaging scenarios.

Figure 6 depicts representative failure cases for each modality, highlighting typical challenges encountered by both our method and the baseline. Among these, the most common failure type is the segmentation of tubular structures such as blood vessels, which are difficult to delineate due to their thin, elongated shapes and often ambiguous boundaries. Notably, in the Ultrasound example, our model achieves a Dice score of 0.88, demonstrating that even in failure cases, the segmentation remains clinically meaningful and outperforms the baseline. These qualitative results reveal the limitations of current approaches in dealing with complex anatomical structures, pointing towards the need for further methodological advances—such as enhanced attention mechanisms or topology-aware losses—to better address these challenging cases.

### 4.3 Ablation Studies

During the development of our method, we carry our some ablation studies to verify whether the proposed strategies improve the scores. Due to the excessive time required by training and evaluation, we only train the model by 12 epochs and evaluate the model with DSC final score on 5% validation set for ablation studies. In each ablation group, we keep all the other situation the same and switch between two strategies for comparison. The results are as follows: **1)** Our simulation strategy for global RoI improves the score from 0.7182 to 0.7361, by

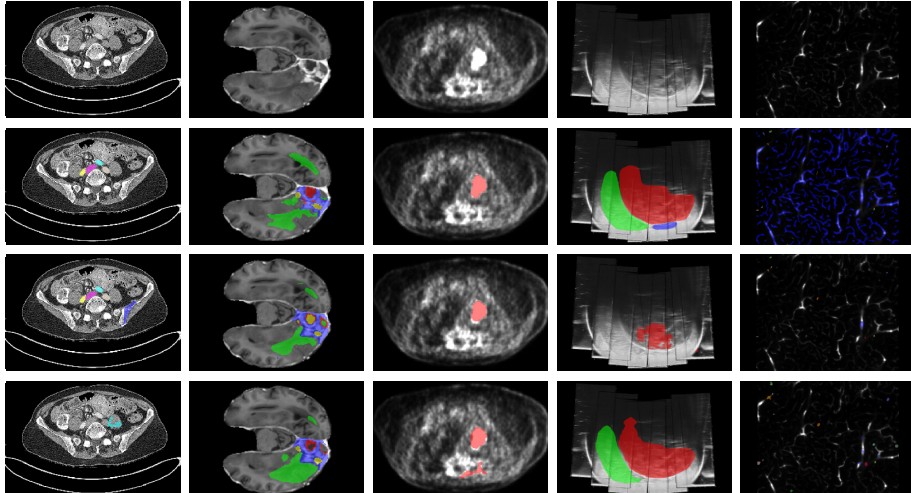

**Fig. 6.** Failed cases on five modalities. Top-to-bottom: Input images, Ground truths, Results of VISTA3D, and Results of ours. Left-to-right: CT, MRI, PET, Ultrasound, and Microscopy.

1.79%. **2)** Our simulation strategy for local RoI improves the score from 0.7124 to 0.7274, by 1.50%. **3)** Using all available prompts in global RoI instead of using the last prompt improves the score by 3.09%. **4)** Generating six points for each interaction in global RoI improves the score by 0.38%.

### 4.4    Results on Final Testing Set

Tables 5 and 6 summarize the quantitative performance of our method compared to four state-of-the-art (SOTA) baselines, including SAM-Med3D [15], VISTA3D [5], SegVol [3], and nnInteractive [4], across five imaging modalities: CT, MRI, Microscopy, PET, and Ultrasound. The testing set images and corresponding labels were provided by the competition organizers, ensuring that no fine-tuning on the test data was possible. This setup guarantees the fairness and reliability of the results.

Consistent with the conclusions drawn from the validation set, our method outperforms the baseline approaches across all imaging modalities. On the coreset, our method achieves DSC_final of 0.8533, 0.6880, 0.6003, 0.7864, and 0.9385 for CT, MRI, Microscopy, PET, and Ultrasound, respectively. Similarly, on the entire dataset, our method achieves DSC_final of 0.8462, 0.6809, 0.5871, 0.7691, and 0.9299 for the respective modalities.

### 4.5    Limitation and Future Work

In this work, we propose a dual-expert mechanism that leverages both global ROI and local ROI branches to handle different segmentation scenarios. However, the

**Table 5.** Quantitative evaluation results of the test set on the **coreset track**.

| Modality | Methods | DSC AUC | NSD AUC | DSC Final | NSD Final |
|---|---|---|---|---|---|
| CT | SAM-Med3D | 2.1225 | 1.7479 | 0.5424 | 0.4478 |
| | VISTA3D | 2.2211 | 1.9899 | 0.5840 | 0.5294 |
| | SegVol | 2.2736 | 2.1592 | 0.5684 | 0.5398 |
| | DCM (ours) | 2.7027 | 2.4484 | 0.6963 | 0.6346 |
| MRI | SAM-Med3D | 2.1169 | 2.1185 | 0.5355 | 0.5371 |
| | VISTA3D | 2.4102 | 2.5065 | 0.6327 | 0.6690 |
| | SegVol | 2.7383 | 3.0322 | 0.6846 | 0.7581 |
| | DCM (ours) | 3.0569 | 3.3125 | 0.7819 | 0.8487 |
| Microscopy | SAM-Med3D | 0.3123 | 0.0311 | 0.0781 | 0.0078 |
| | VISTA3D | 1.8416 | 2.5043 | 0.4701 | 0.6285 |
| | SegVol | 2.7359 | 3.8765 | 0.6840 | 0.9691 |
| | DCM (ours) | 3.1926 | 3.9722 | 0.8092 | 0.9943 |
| PET | SAM-Med3D | 2.3133 | 1.7599 | 0.5826 | 0.4465 |
| | VISTA3D | 1.7576 | 1.2811 | 0.4539 | 0.3332 |
| | SegVol | 2.7847 | 2.2974 | 0.6962 | 0.5744 |
| | DCM (ours) | 3.0282 | 2.6575 | 0.7730 | 0.6907 |
| Ultrasound | SAM-Med3D | 0.6009 | 0.4005 | 0.1502 | 0.1001 |
| | VISTA3D | 0.7571 | 0.8212 | 0.2378 | 0.3116 |
| | SegVol | 0.7931 | 1.2851 | 0.1983 | 0.3213 |
| | DCM (ours) | 1.5276 | 1.9049 | 0.4500 | 0.5894 |

current design of the local ROI branch is not yet optimal. First, although our balanced prompt sampling strategy for local ROI is an improvement over the purely random approach used in VISTA3D, it still does not fully reflect real interactive behaviors. Second, the local ROI branch is intended for use cases with only point-based prompts, and thus, training should ideally be restricted to samples with point-only guidance. In practice, however, we train this branch on the entire dataset regardless of the prompt type, which is suboptimal.

Additionally, comparative analysis of Table 3 and Table 4 reveals another limitation of our current approach. When moving from the coreset track to the all-data track, baseline methods such as SAM-Med3D, VISTA3D, and SegVol exhibit clear performance improvements across most metrics. In contrast, our method does not show a similarly significant improvement. This suggests that the local ROI branch, in its current form, may not be fully leveraging the advantages of larger datasets, possibly due to its design or training strategy.

Future work will focus on several directions: **1)** Developing more realistic and user-centric prompt sampling strategies for interactive point generation in the local ROI branch. **2)** Restricting local ROI branch training to point-prompt samples, potentially through a tailored data loader or curriculum learning to better match the intended deployment scenario. **3)** Exploring advanced fusion mechanisms between global and local branches to further enhance segmentation accuracy, especially in challenging cases. These improvements are expected to

**Table 6.** Quantitative evaluation results of the test set on the **all-data track**.

| Modality | Methods | DSC AUC | NSD AUC | DSC Final | NSD Final |
|---|---|---|---|---|---|
| CT | SAM-Med3D | 2.1937 | 1.7846 | 0.5711 | 0.4672 |
| | VISTA3D | 2.3482 | 2.1062 | 0.6198 | 0.5616 |
| | SegVol | 2.4358 | 2.3213 | 0.6089 | 0.5803 |
| | nnInteractive | 3.1831 | 3.1286 | 0.8342 | 0.8355 |
| | DCM (ours) | 2.8017 | 2.5760 | 0.7231 | 0.6734 |
| MRI | SAM-Med3D | 2.1064 | 2.0427 | 0.5317 | 0.5169 |
| | VISTA3D | 2.4891 | 2.5825 | 0.6516 | 0.6859 |
| | SegVol | 2.8377 | 3.1261 | 0.7094 | 0.7815 |
| | nnInteractive | 3.3866 | 3.6611 | 0.8680 | 0.9416 |
| | DCM (ours) | 3.1327 | 3.4052 | 0.8022 | 0.8730 |
| Microscopy | SAM-Med3D | 0.3115 | 0.1726 | 0.0778 | 0.0431 |
| | VISTA3D | 2.4526 | 3.4035 | 0.6231 | 0.8528 |
| | SegVol | 2.9603 | 3.9472 | 0.7401 | 0.9868 |
| | nnInteractive | 3.4580 | 3.9895 | 0.8743 | 0.9980 |
| | DCM (ours) | 3.3095 | 3.9834 | 0.8358 | 0.9963 |
| PET | SAM-Med3D | 1.3004 | 0.7297 | 0.3285 | 0.1844 |
| | VISTA3D | 1.8687 | 1.3919 | 0.4688 | 0.3523 |
| | SegVol | 2.9844 | 2.5108 | 0.7461 | 0.6277 |
| | nnInteractive | 3.2230 | 3.0753 | 0.8170 | 0.7854 |
| | DCM (ours) | 3.1477 | 2.8841 | 0.7989 | 0.7459 |
| Ultrasound | SAM-Med3D | 0.8313 | 0.7004 | 0.2078 | 0.1751 |
| | VISTA3D | 0.9072 | 1.2257 | 0.2953 | 0.4789 |
| | SegVol | 0.9429 | 1.4435 | 0.2357 | 0.3609 |
| | nnInteractive | 2.4088 | 3.0407 | 0.7073 | 0.8886 |
| | DCM (ours) | 1.7710 | 2.2812 | 0.5144 | 0.6767 |

further bridge the gap between automated training and real-world interactive segmentation.

## 5    Conclusion

In this paper, we presented DCM, a novel interactive segmentation framework that integrates both global and local RoI strategies within a dual-expert architecture. By combining a global-RoI expert for capturing overall anatomical context and a local-RoI expert for handling fine structures and limited prompts, our method addresses key challenges in 3D medical image segmentation. The proposed interaction simulation strategies further improve the training efficiency and alignment with real-world user behaviors. Extensive quantitative experiments on five imaging modalities demonstrate clear performance improvements over leading baselines. Specifically, on the coreset track, our DCM improves the final DSC by 29.43% on CT, 29.85% on MRI, 52.35% on Microscopy, 25.20% on PET, and 55.44% on Ultrasound relative to SAM-Med3D. Compared to VISTA3D, the Dice improvements are 13.86% (CT), 11.03% (MRI), 15.48% (Microscopy), 17.41% (PET), and 23.11% (Ultrasound), respectively. These substan-

tial gains are also reflected in boundary accuracy and are consistently observed on the all-data track. While there remain challenges in accurately segmenting thin or irregular structures, our results highlight the potential of dual-expert designs for advancing interactive medical image analysis. Future work will explore more realistic prompt simulation, tailored training for local RoI, and advanced fusion between expert branches to further enhance segmentation performance and applicability in clinical workflows.

**Acknowledgements.** We thank all the data owners for making the medical images publicly available and CodaLab [17] for hosting the challenge platform.

**Disclosure of Interests.** The authors have no competing interests to declare that are relevant to the content of this article.

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

**Table 7.** Checklist Table. Please fill out this checklist table in the answer column. (**Delete this Table in the camera-ready submission**)

| Requirements | Answer |
| --- | --- |
| A meaningful title | Yes |
| The number of authors ($\leq$6) | 3 |
| Author affiliations and ORCID | Yes |
| Corresponding author email is presented | Yes |
| Validation scores are presented in the abstract | Yes |
| Introduction includes at least three parts: background, related work, and motivation | Yes |
| A pipeline/network figure is provided | Figure 1 |
| Pre-processing | Page 7 |
| Strategies to data augmentation | Page 8 |
| Strategies to improve model inference | Unused |
| Post-processing | Unused |
| Environment setting table is provided | Table 1 |
| Training protocol table is provided | Table 2 |
| Ablation study | Page 11 |
| Efficiency evaluation results are provided | Table 2 |
| Visualized segmentation example is provided | Figures 5 & 6 |
| Limitation and future work are presented | Yes |
| Reference format is consistent. | Yes |
| Main text $>=$ 8 pages (not include references and appendix) | Yes |