# OpenReview forum: "Rethinking RoI Strategy in Interactive 3D Segmentation for Medical Images"
_thecvf.com/CVPR/2025/Workshop/MedSegFM — CVPR 2025 Workshop MedSegFM Submission_

### Official Review · Reviewer_TQ2t · 2025-09-16
**Rethinking RoI Strategy in Interactive 3D Segmentation for Medical Images**

**Rating:** 8
**Confidence:** 5

**Review:**

1.Summary
This paper proposes a dual-expert framework named DCM (DualClickMed) for interactive 3D medical image segmentation. Its core innovation lies in the parallel use of two complementary Region-of-Interest (RoI) strategies: a "global-RoI expert" leverages all user interaction points to understand the overall anatomical context of an organ, while a "local-RoI expert" focuses on high-resolution patches around the final interaction point to achieve fine-grained segmentation. To enhance model performance, the authors also designed interaction simulation strategies for both experts that more closely align with real-world clinical workflows. Extensive experiments on a challenge dataset spanning five modalities demonstrate that the proposed method outperforms current state-of-the-art approaches in both segmentation accuracy and boundary agreement.

2.Strengths

1）The decoupling of global context and local detail, handled by a dual-expert model, is an elegant and effective design. This architecture directly addresses the inherent conflict between "seeing the big picture" and "refining the details" in 3D medical imaging, showing high potential for clinical application.

2）The proposed interaction simulation strategies (boundary-attenuated and balanced sampling) are another major highlight. The work moves beyond the overly simplistic random sampling of previous methods to instead emulate the annotation behavior of human experts. This makes the model training more targeted and significantly improves its ability to handle complex and ambiguous boundaries.

3）The paper provides a thorough evaluation on a public dataset containing multiple modalities (CT, MRI, PET, etc.), comparing against several state-of-the-art models on both the coreset and all-data tracks. The results consistently demonstrate the superiority of the proposed method, providing strong evidence of its generalizability and robustness.

4）The paper is well-structured and logically sound. The presentation, from problem formulation and method design to experimental analysis, is exceptionally clear. Furthermore, the authors' candid discussion of the method's limitations reflects a rigorous scientific attitude.

3. Weaknesses

1）The current coordination mechanism between the global and local experts is too simplistic. First, the switching rule, based solely on whether "only point prompts are provided," is somewhat arbitrary and may fail in complex interaction scenarios. Second, there is a mismatch between the training data for the local expert and its intended use case (point-only prompts), which could limit its optimal performance under those specific conditions.

2）As seen in the results (Table 3 vs. Table 4), when scaling from the coreset to the all-data track, the performance gain of DCM is less pronounced than that of baselines like nnInteractive. This suggests that the model, particularly the local expert branch, may not be fully capitalizing on the benefits of larger-scale data, and its scalability could be improved.

3）Have you considered enabling a deeper information exchange between the global and local experts, rather than a simple binary switch? For example, could the global expert provide a probabilistic prior map to guide the local expert's fine-grained segmentation, thus creating a coarse-to-fine collaborative workflow?

4）Observing the failure cases in Figure 6, the main difficulties appear in segmenting thin, tubular structures like blood vessels. Do you attribute this primarily to the RoI strategy itself (e.g., the global RoI resolution being insufficient to capture fine structures) or to the model architecture (e.g., standard convolutions or transformers struggling to effectively model topological connectivity)?

---

> ### Author Rebuttal · Authors · 2025-11-05
>
> We are grateful to the reviewer for the insightful and constructive feedback, which has been instrumental in improving the overall clarity and organization of our manuscript. We have thoroughly considered the comments and revised the manuscript accordingly.
>
> 1. The current method for selecting between the global and local experts is relatively simplistic and somewhat arbitrary. Our current approach was specifically designed to fit the data and requirements of this competition. For a more generalizable solution, we plan to develop more sophisticated strategies for coordinating the global and local experts in future work, rather than relying solely on the presence or absence of a box prompt.
>
> 2. This is indeed the case. As you pointed out, our method performs less effectively on thin and tubular structures like blood vessels, which highlights an area that requires improvement. nnInteractive does outperform our approach in this aspect, and we are actively investigating how it handles such structures to better understand and address the limitations of our model, particularly in the local expert branch.
>
> 3. We have considered deeper integration or a cascaded approach to enable better collaboration between the global and local experts. For example, using the global expert to provide a probabilistic prior map to guide the local expert’s fine-grained segmentation is a promising idea. However, such methods typically introduce additional computational overhead, which is a concern given the real-time efficiency requirements of this competition. Balancing the trade-off between performance and computational efficiency is critical, and this is something we plan to explore further in our future work.
>
> 4. Segmenting thin and tubular structures like blood vessels is indeed a challenge. In this competition, the difficulty is further compounded by the fact that no boxes are provided for such structures, making it challenging to accurately localize them. As a result, the subsequent point refinement step struggles to effectively capture their complex topological connectivity. Addressing this limitation and improving the segmentation of such structures will be a key focus of our future work.
>
> We sincerely thank you once again for your insightful suggestion, which provides valuable direction for improving our work in the future.

---

### Official Review · Reviewer_CVPN · 2025-09-26
**Simple and effective strategy design of global-> local refinement**

**Rating:** 8
**Confidence:** 4

**Review:**

The paper proposes DualClickMed (DCM), a dual-expert interactive segmentation framework for 3D biomedical images. The main idea is to combine a global ROI expert for holistic anatomical context with a local ROI expert for fine-grained segmentation details. Prompt simulation strategies was also designed to better mimic real user interactions during training. Good evaluation results on five modalities (CT, MRI, PET, Ultrasound, Microscopy) using the CVPR 2025 Interactive 3D Biomedical Segmentation Challenge dataset over the baselines (SAM-Med3D, VISTA3D, SegVol, and nnInteractive).

- Strength:
  - The dual-expert design is effective in balancing global context and local details, addressing the limitations of prior approaches, especially in the 3D context.
  - The boundary-attenuated sampling and balanced prompt simulation was shown to improve the performance, which aligns better with human interaction.
  - Results span multiple imaging modalities on both coreset and all-data tracks showed significant improvements over baseline models, especially on the coreset track, and for ultrasound segmentation.
  - Ablation study was done for simulation strategies and prompt sampling, showing effective of he methods.

- Weaknesses (and questions):
  - The model architecture in Figure 1 could be a little confusing, as the global ROI and local ROI seem to aim for different anatomies (kidney and pancreas), and the final mask output is a multiclass one. Not sure if this was by design.
  - The training objective was simply stated in terms of standard segmentation loss functions. However, due to the design of DCM involving global and local ROI expert, it would be better to explain how the two modules was trained in more details.
  - For the global ROI module with ViT, the input patch embedding was reduced from 16x16x16 to 8x8x8, which indeed improved the computational efficiency. However, with the same resolution, if you reduce the number of patches instead of patch dimension, you should see significantly more gain in efficiency, as the transformer's self attention computational complexity is $O(n^2d)$, so with the same resolution budget reducing n is quadratically more efficient.
  - The performance on all-data track does not outperform the best baseline, but still outperforming the baselines in the coreset, except nnInteractive.

---

> ### Author Rebuttal · Authors · 2025-11-05
>
> We sincerely appreciate the reviewer’s thoughtful and detailed comments, which have greatly contributed to enhancing the quality and presentation of our manuscript. We have carefully addressed these suggestions and made corresponding revisions in the manuscript.
> 1. Our strategy is not determined by different anatomies but rather by the type of prompt, which dictates whether the global or local module is used. In Figure 1, different organs were combined into a single illustration for better visual representation. We have revised Figure 1 accordingly to make this clearer.
>
> 2. Thank you for pointing this out. In our method, the global and local RoI modules are trained separately based on the type of prompt provided. Specifically, for data with box prompts, we train the global RoI module, while for data without box prompts, we train the local RoI module. Aside from the initial prompt type, the remaining refinement process is identical for both modules. We have clarified this training strategy in the manuscript.
>
> 3. We acknowledge the suggestion of reducing the number of patches instead of the patch embedding dimensions to achieve greater computational efficiency due to the quadratic complexity of the transformer's self-attention mechanism. However, we chose to reduce the patch embedding dimensions because our model is initialized with the pre-trained weights from MedSAM-3D. By doing so, except for the first layer, the parameters of the other layers can be directly loaded from the pre-trained MedSAM-3D, which not only allows us to achieve better performance but also ensures better convergence and stability during training. We will consider incorporating your proposed approach in future work, in combination with our current strategy to further improve efficiency.
>
> 4. Our method indeed does not perform as well as nnInteractive; however, it still outperforms other baseline methods by a significant margin. Compared to nnInteractive, our method has an advantage in terms of training efficiency, as we can leverage the pre-trained MedSAM-3D checkpoint to initialize our model. We are actively analyzing the differences between our approach and nnInteractive, and we believe that optimizing our method for tubular structures like blood vessels will further enhance its performance.
>
> We sincerely thank you once again for your thoughtful and constructive feedback, which has been invaluable in enhancing the clarity, quality, and overall rigor of our work.

---

### Official Review · Reviewer_sQzP · 2025-10-11
**Clear Concept but Incomplete Justification of Design Choices and Evaluation**

**Rating:** 8
**Confidence:** 5

**Review:**

### Summary
The paper proposes a new region of interest (RoI) strategy for interactive 3D medical image segmentation. It combines a Global RoI model that captures whole organ context with a Local RoI model focused on fine details. The authors also explore a new way to simulate user prompts to make the training process more realistic. The approach is evaluated on multiple organs and shows good results on a core dataset compared to existing methods.

### Strengths
1. The study explores a novel simulation of interactive prompts that better mimics real user behavior. This is important because current interactive models often fail to perform consistently across different organs and tissue types when deployed in real clinical settings.
2. The performance on the core set is strong and clearly demonstrates the effectiveness of the proposed approach compared to other existing methods.
3. The two-branch design combining Global and Local RoI processing is an interesting idea that attempts to balance contextual understanding with fine-detail segmentation.
4. The paper is well written and does a thorough job comparing to other benchmark models.

### Weaknesses
1. Several figures are confusing. In Figure 1, multiple organs are segmented simultaneously, which is not typical for interactive segmentation, and prompt colors are unclear. Figures 2 and 3 also lack clear labeling of what each color and shape represents.
2. The Global RoI section assumes that the entire organ can be captured from limited user prompts, but this depends heavily on how those prompts are distributed. This conditional limitation should be clearly acknowledged.
3. The Local RoI model choice (SegResNet) is not justified, especially since the Global RoI model uses a different backbone. A short explanation for this difference is needed.
4. The scaling factor of 1.8 for RoI expansion is mentioned without explaining how it was derived or under what conditions it holds true.
5. The limitations section attributes lack of performance improvement to the Local RoI branch without clear supporting evidence. This conclusion should be better justified or labeled as speculation. An additional ablation experiment comparing inference with only the Global branch, only the Local branch, and the combined switching strategy would provide valuable insight into each component’s contribution.

---

> ### Author Rebuttal · Authors · 2025-11-05
>
> We thank the reviewer for the valuable and constructive feedback, which helped us improve the clarity, completeness, and presentation of our manuscript. We have carefully considered the suggestions and revised the manuscript accordingly:
>
> 1. Our method follows the typical interactive segmentation approach, where one organ is segmented at a time. Figure 1 has been revised to focus on a single organ for clearer demonstration. Additionally, we have modified Figures 1, 2, and 3 to clarify prompt colors and include proper labeling for better understanding.
>
> 2. It's correct that the performance of the Global RoI module relies on the assumption that the entire organ can be captured based on the provided input prompts. The Global RoI module assumes the entire organ can be captured based on the provided prompts, and this is valid under the competition’s setting, where the provided boxes are designed to include the complete organ. Additionally, we applied a scaling factor during preprocessing to ensure full coverage of the organ. This clarification has been added to the manuscript.
>
> 3. The Global and Local RoI modules serve different purposes, which is why we chose different network architectures. The Local RoI module adopts the same backbone as in ViTSA-3D, as it offers better local feature representation. We have added an explanation in the manuscript to justify this design choice.
>
> 4. The scaling factor of 1.8 was chosen based on the observation that it is generally sufficient to include the whole organ. In MedSAM-3D, one major failure case involved larger organs not being fully covered, leading to suboptimal performance. To address this, we adjusted the RoI box size to ensure full coverage and found 1.8 to be effective. This explanation has been clarified in the manuscript.
>
> 5. Thank you for the suggestion. We will include ablation experiment results in the final version to analyze the contributions of each branch, including inference with only the Global branch, only the Local branch, and the combined switching strategy.
>
> Thank you again for your insightful feedback, which has been invaluable in helping us improve our work.

---

### Decision · Program_Chairs · 2025-11-12

Accept